# Serine/Threonine Protein Phosphatases 1 and 2A in Lung Endothelial Barrier Regulation

**DOI:** 10.3390/biomedicines11061638

**Published:** 2023-06-05

**Authors:** Rahul S. Patil, Anita Kovacs-Kasa, Boris A. Gorshkov, David J. R. Fulton, Yunchao Su, Robert K. Batori, Alexander D. Verin

**Affiliations:** 1Vascular Biology Center, Medical College of Georgia, Augusta University, Augusta, GA 30912, USA; 2Department of Pharmacology, Medical College of Georgia, Augusta University, Augusta, GA 30912, USA

**Keywords:** endothelial cells, cytoskeleton, permeability, Ser/Thr protein phosphatase 1, Ser/Thr protein phosphatase 2A, phosphorylation, inflammation

## Abstract

Vascular barrier dysfunction is characterized by increased permeability and inflammation of endothelial cells (ECs), which are prominent features of acute lung injury (ALI), acute respiratory distress syndrome (ARDS), and sepsis, and a major complication of the SARS-CoV-2 infection and COVID-19. Functional impairment of the EC barrier and accompanying inflammation arises due to microbial toxins and from white blood cells of the lung as part of a defensive action against pathogens, ischemia-reperfusion or blood product transfusions, and aspiration syndromes-based injury. A loss of barrier function results in the excessive movement of fluid and macromolecules from the vasculature into the interstitium and alveolae resulting in pulmonary edema and collapse of the architecture and function of the lungs, and eventually culminates in respiratory failure. Therefore, EC barrier integrity, which is heavily dependent on cytoskeletal elements (mainly actin filaments, microtubules (MTs), cell-matrix focal adhesions, and intercellular junctions) to maintain cellular contacts, is a critical requirement for the preservation of lung function. EC cytoskeletal remodeling is regulated, at least in part, by Ser/Thr phosphorylation/dephosphorylation of key cytoskeletal proteins. While a large body of literature describes the role of phosphorylation of cytoskeletal proteins on Ser/Thr residues in the context of EC barrier regulation, the role of Ser/Thr dephosphorylation catalyzed by Ser/Thr protein phosphatases (PPases) in EC barrier regulation is less documented. Ser/Thr PPases have been proposed to act as a counter-regulatory mechanism that preserves the EC barrier and opposes EC contraction. Despite the importance of PPases, our knowledge of the catalytic and regulatory subunits involved, as well as their cellular targets, is limited and under-appreciated. Therefore, the goal of this review is to discuss the role of Ser/Thr PPases in the regulation of lung EC cytoskeleton and permeability with special emphasis on the role of protein phosphatase 1 (PP1) and protein phosphatase 2A (PP2A) as major mammalian Ser/Thr PPases. Importantly, we integrate the role of PPases with the structural dynamics of the cytoskeleton and signaling cascades that regulate endothelial cell permeability and inflammation.

## 1. Introduction

The vascular endothelium is crucial to maintaining blood vessel wall homeostasis. In addition to regulating vasomotion and thrombosis, the endothelium acts as a semi-selective diffusion barrier between the plasma and interstitial fluid [1,2,3]. Inflammatory mediators, such as histamine, thrombin, lipopolysaccharide (LPS), and cytokines stimulate multiple intracellular signaling pathways by interacting with their cognate receptors on the endothelial cell (EC) surface. These inflammatory signaling pathways result in the formation of paracellular gaps, which ultimately increase trans-endothelial permeability for fluid and macromolecules [1,4,5,6]. It is well established that contractile and tethering forces are the primary regulators of the EC paracellular barrier [4,5]. While Karki and Birukova recently credited microtubules as the chief regulator of EC barrier function, based on data in lung injury [7], a large body of literature has demonstrated critical roles for other cytoskeletal components such as intercellular junctions and microfilaments (actin and myosin) in structural as well as functional changes in the EC barrier (reviewed in [2,4,5,8]). More broadly, it is the combination of microfilament proteins, the degree of myosin light chain (MLC) phosphorylation, and the number of intercellular junctions determining the contraction and paracellular integrity, and thus, the barrier function/dysfunction of ECs [6,8,9,10].

Dysfunctional remodeling of the EC cytoskeleton can provoke Acute Lung Injury (ALI), its more severe manifestation Acute Respiratory Distress Syndrome (ARDS), and other cardiovascular diseases. The recent review by Matthay et al. provides a thorough overview of the existing pathophysiological mechanisms, revision of diagnostic paradigms, and future directions on ARDS [11]. ALI may result from infection, sepsis, aspiration, multiple events of trauma or shock, and other insults eliciting inflammation, membrane injury to the alveolar-capillary (compromise of endothelial and epithelial barriers), and increased permeability of pulmonary edema, which may lead to pulmonary fibrosis and pulmonary vascular rarefaction and may turn, as a result, into multiple organ failure [12]. Importantly, a growing body of evidence directly implicates the loss of EC barrier integrity as a major cause of COVID-19 complications (C-ARDS) [13,14].

It is well accepted that the cytoskeletal architecture of ECs is regulated by post-translational modification of cytoskeletal proteins such as reversible phosphorylation/dephosphorylation [15,16,17]. Phosphorylation is a major post-translational modification regulating many cellular functions and approximately one-third of all human cellular proteins undergo reversible phosphorylation on either Ser, Thr, or Tyr residues [18]. The human genome encodes 518 putative kinases [19], which are responsible for adding phosphates, and 189 known or predicted phosphatases [20] that remove phosphate groups. The role of Ser/Thr protein kinases, including MLC kinase (MLCK), Rho kinase, protein kinase C (PKC), and mitogen-activated protein kinases (MAP kinases) as key regulators of cytoskeletal and junctional rearrangements that impair lung EC barrier function has been well characterized [17,21,22,23,24,25]. In contrast, other Ser/Thr kinases such as protein kinase A (PKA), AMP-activated protein kinase (AMPK), and p21-activated kinase (PAK) promote EC barrier protection [17,26,27,28,29]. Less well appreciated are the roles of Ser/Thr phosphatases in lung EC cytoskeletal remodeling and barrier regulation, due, at least in part, to the complexity of the PPase holoenzyme structure, which is comprised of conserved catalytic subunits and many variable regulatory (targeting) subunits that determine substrate specificity [30,31].

Ser/Thr phosphoprotein phosphatases such as PP1 and PP2A account for the majority of cellular Ser/Thr dephosphorylation events [32,33] and play critical roles in the preservation of the lung EC barrier [34,35,36,37]. However, the mechanisms underlying the ability of PP1/PP2A to strengthen the EC barrier function are complex and not completely understood. In this review, we will summarize current knowledge of the multifaceted involvement of Ser/Thr phosphatases 1 and 2A in the regulation of lung EC permeability, focusing on their roles in cytoskeletal arrangement (Figure 1).

## 2. Ser/Thr-Specific Protein Phosphatases, an Overview and Classification

The majority of cellular signaling activities are regulated via reversible protein phosphorylation driven by the opposing action of kinases and phosphatases [30]. Proteomic analysis revealed that more than 98% of human protein phosphorylation occurs either on Ser or Thr residues [38]. A broad implication of this is that Ser/Thr phosphorylation/dephosphorylation are major protein modifications that can acutely impact many aspects of protein function.

Initially, Ser/Thr protein phosphatases were subdivided into four groups (1, 2A, 2B, and 2C) based on their substrate specificity, divalent cation requirements, and susceptibility to selective PPase inhibitors [39]. Further cloning and functional characterization studies changed the classification of the Ser/Thr PPase family, which is now recognized to contain three subfamilies: phosphoprotein PPases (PPPs), metal-dependent protein PPases (PPMs), and aspartate-based PPases (FCP/SCP) [30]. According to this classification, PP1 and PP2A belong to the PPP family, which is characterized by a multi-subunit structure comprised of a conserved core catalytic subunit that has a relatively broad substrate specificity that can be attributed to a variety of regulatory subunits, which provided a high level of versatility [39,40,41] (Figure 1). In most cells, PP1 and 2A are the most abundant Ser/Thr phosphatases and control a variety of cellular functions, including the endothelial barrier [32,33,40].

## 3. Ser/Thr Protein Phosphatase 1

### 3.1. Structure-Functional Dynamics

PP1 is a major phosphatase in eukaryotes and has been implicated in cellular functions such as cell division, glycogen metabolism, contraction of muscle, etc. (reviewed in [42,43]). Initial structure-functional studies revealed that most of the mammalian PP1 holoenzymes are dimers of α, β/δ, γ1, or γ2, isoforms of catalytic subunit 1 (PP1_C_ or CS1), which are encoded by three independent genes (with two splice variants for γ) and a regulatory (R) subunit [44,45,46] (Figure 1).

The crystal structure of various PP1_C_ isoforms shares a similar structure comprised of two firmly interlinked N-terminal and C-terminal domains with three distinct and attached clefts [47,48,49,50]. PP1_C_ shows a compact α/β fold, in which a β sandwich is embedded within two α-helical domains. The location of the catalytic site is centralized to three shallow furrows, molding it into a Y-shape. Mn^2+^ and Fe^2+^ are metal ions that are present in the catalytic site, coordinated by one asparagine, two aspartic acid, and three histidine residues, which are the most conserved amino acids among all members of the PPP family. These ions activate an H_2_O molecule, which triggers a nucleophilic attack on the phosphorous; hence, they bear special recognition in the catalytic process [51]. Toxins such as okadaic acid [52] and microcystin [53] are closely integrated with the catalytic site via interactions with its outer residues and sterically inhibit PP1_C_ activity. The unaccompanied existence of PP1_C_ subunits is rare in cells. The overexpression of yeast PP1_C_, GLC7, was shown to be detrimental almost 3 decades ago [54,55] and reflects the need for regulatory subunits (proteins) to firmly control PP1_C_ activity in cells.

Many articles have been published on the structural basis of its regulation as well as the specificity of PP1 [49,50,56,57]. While PP1_C_ alone has broad substrate specificity, the variety of the R subunits provides for specificity, localization, and temporal regulation of the PP1 holoenzymes [58,59]. Every R subunit differently impacts the function and regulation of PP1. For example, PP1 localized to the nucleus has a different substrate profile mostly based on the regulatory subunit responsible for nuclear targeting. In another example, the holoenzyme PP1-inhibitor 2 complex controls the degree of phosphorylation of proteins that participate in cell-cycle control [60], the PP1-nuclear inhibitor of protein phosphatase 1 (NIPP1) complex dephosphorylates many splicing factors [61], etc.

Although there are no apparent similarities in the primary structures of R subunits, they share multiple short interaction sites [58,59]. Most of the R subunits have an RVxF docking motif, which is a unique requirement for interactions with the PP1 catalytic subunit [62]. Interaction of the R subunits with PP1_C_ via RVxF motif does not affect the activity of PP1_C_ as the interaction site is distant from the catalytic site on PP1_C_ [49,56]. In addition, the PP1_C_ β/δ isoform (aa 301–309) contains a C-terminal region that interacts with the ankyrin repeats of some regulatory subunits [49]. Further, five unique residues in the PP1_C_ β/δ isoform central region are responsible for specific interaction with the R subunits holding MyPhone (myosin phosphatase N-terminal element) binding motif (RxxQ[VIL][KR]x[YW]) at the N-terminus [62,63,64,65].

Interestingly, despite the validation of a large number of PP1_C_-binding proteins, only a few have been evolutionarily conserved in eukaryotes [43]. Structure-function analysis of regulatory subunits reveals that some regulatory subunits are comprised of intrinsically disordered regions (IDR). In the unbound form, they have a flexible structure allowing them to form unique interactions with PP1_C_ via multiple docking motifs [64]. The occurrence of Short Linear Motif/s (SLiM) indicates a common feature of the IDR by which various unstructured R subunits can bind to the PP1_C_ [43].

A large number of articles have been published in the last two decades, accelerating our knowledge of protein phosphatase interactors and structures [30,43,66,67,68]. Based on advanced proteomic approaches, it was suggested that PP1 forms a stable complex with more than 650 proteins (PP1-interacting proteins, PIPs; re-named lately to regulatory interactors of protein phosphatase 1 (RIPPOs) [69] and a concept of PP1 protein interactome was introduced, in which interactome includes numerous RIPPOs with multiple substrate-targeting domains [41,64]. Heroes et al. interpreted that PP1 interactome diversity, as well as features of the PP1 binding code, determine the magnificent specificity of PP1 [41].

### 3.2. PP1, Cytoskeleton, and EC Barrier Regulation

The weakening of barrier function in pulmonary ECs upon inhibition of Ser/Thr protein phosphatases [70,71] was first shown more than 2 decades ago. Figure 2A demonstrated that Ser/Thr PPase inhibitor, calyculin A (inhibits both PP1 and 2A) [72] increases human lung microvascular EC permeability (decreases transendothelial electrical resistance (TER), an inverse permeability index [73]. Further, calyculin and tautomycetin (inhibits PP1) [74] increase lung vascular leak in mice (Figure 2B). Early reports on the role of PP1 in pulmonary EC barrier regulation demonstrated that EC barrier dysfunction induced by PP1 inhibitors is tightly linked to an increase in MLC phosphorylation, indicating the involvement of MLC phosphatase (MLCP) in EC barrier protection [69]. Partial purification and subsequent cloning of EC MLCP demonstrated that, similar to smooth muscle (SM) MLCP, EC MLCP is a PP1 enzyme, which comprises PP1_C_ β/δ isoform and Myosin phosphatase targeting subunit 1 (MYPT1) [70,75,76,77]. Human EC MYPT1 is composed of two isoforms, long and variant 2 (V2), lacking 56 amino acids from the central part of human MYPT1 long (Figure 3). They correspond to variants 1 and 4 (rat SM) [77] and long and variant 2 (V2) (HeLa cells) [78]. Human MYPT1 (PPP1R12A; GenBank ID#219842212) and its spliced variants are encoded by one single gene on human chromosome 12q15-q21.2 [79]. Similar to its SM counterpart, EC MYPT1 possesses PP1_C_ binding motif, Lys-Val-Lys-Phe (^35^KVKF) motif, and eight-repeat ankyrin motif (involved in substrate binding) at the N-terminus [49]. While the binding of short 1–300 aa N-terminal fragment (“constitutively active”, C/A fragment) of MYPT1 is sufficient to allosterically regulate PP1 activity [80], the MYPT1 C-terminal part with the Leucine-zipper motif is primarily responsible for direct substrate binding [81,82] (Figure 3). In contrast to SM, M20, a small MLCP subunit with an unknown function, was not found in ECs [35,75,83]. Experiments with down-regulation of MLCP (either MYPT1 or PP1_C_) demonstrated that EC MLCP is directly involved in EC barrier regulation in vitro and in vivo [34,35]. Similar to its SM counterpart, the EC MYPT1 function regulates by inhibitory Rho-dependent phosphorylation at Thr696 or Thr853 [84,85,86,87]. The EC barrier compromise and increase in MLC phosphorylation are induced by MT inhibitors, which are attributed to a Rho-dependent decrease in MLCP activity, but not to an increase in MLCK activity [85]. The Rho-dependent inhibition of MLCP is involved in EC barrier compromise induced by a variety of edemagenic agents such as thrombin, LPS, Transforming growth factor-beta1 (TGFβ-1), etc. [34,86,88]. It is worth noting that multiple other kinases are able to phosphorylate MYPT1 at Thr696 and/or Thr853 (reviewed in [81,89,90,91]). Some of them, such as zipper-interacting protein kinase (ZIPK), Raf-1, and integrin-linked kinase (ILK), are involved in EC barrier regulation in various EC types [92,93,94]. However, whether they affect the lung microvascular EC barrier via phosphorylation/inhibition of MYPT1 is not clear.

Purinergic agonists such as extracellular ATP increase MLCP activity in pulmonary ECs, which is accompanied by a decrease in MLC phosphorylation and EC barrier strengthening. Further, the depletion of MLCP subunits (either MYPT1 or PP1_C_ β/δ, but not α isoform) attenuates lung EC barrier enhancement induced by ATP (or ATPγS, a slow hydrolyzable analog), or adenosine [35]. While the mechanism of purine-mediated MLCP activation is not entirely clear, it may involve a purine-induced increase in PKA activity, leading to MYPT1 phosphorylation, which may counteract with inhibitory Rho-mediated MYPT1 phosphorylation [95]. Interestingly, that while Gs-mediated adenosine-induced MLC dephosphorylation requires both PKA and direct cAMP effector, EPAC1 (Exchange protein directly activated by cAMP 1), Gi-mediated ATPγS-induced MLC dephosphorylation and barrier enhancement apparently involved interaction of MYPT1 with A-kinase anchoring protein-2 (AKAP2) [26], suggesting complex and agonist-specific involvement of purinergic signaling in MYPT1 activation.

In addition to direct phosphorylation of MYPT1, the regulation of MLCP activity can be achieved via MYPT1 interaction with specific RIPPOSs (formerly PIPs) [96], such as PKC-activated phosphatase inhibitor protein with a molecular weight of 17 kDa (CPI17) [97]. Phosphorylation of CPI17 at Thr38 by PKC exposes CPI17’s inhibitory surface, strengthening the attachment of CPI17 to MLCP and thus producing a more than 1000-fold increase in inhibitory potency [98,99,100]. Unlike other endogenous PPase 1 inhibitors (inhibitors 1 and 2), CPI17 can inhibit the heterotrimeric form of MLCP, without dissociation of MLCP subunits [98]. In smooth muscle, phosphorylated CPI17 selectively inhibits MLCP among other PP1 holoenzymes, leading to an increase in SM contraction [99]. While PKCα and δ isoforms have been identified as enzymes largely responsible for CPI17 phosphorylation leading to histamine-induced vasoconstriction in the aorta [101], the role of other PKC isoforms as well as other kinases, such as Rho kinase, ILK, ZIPK in CPI17 phosphorylation has also been described in SM (reviewed in [102,103]). Indeed, it has been recently proposed that both PKC and Rho kinase, activated sequentially, are responsible for sustained CPI17 phosphorylation and Ca^2+^ sensitization in SM [103].

While CPI17 mRNA is present in human macro and microvascular ECs, the expression of CPI17 in microvascular ECs is more evident [104]. Phorbol myristate acetate (PMA, a direct PKC agonist) dramatically enhances EC contractile machinery (stress fibers and MLC phosphorylation) in a PKCα-dependent, but Rho-independent, manner in human pulmonary artery ECs (HPAECs) with ectopic CPI17 expression [104]. Further, the depletion of endogenous CPI17 in HLMVECs attenuates histamine-, but not thrombin-induced EC barrier compromise [104]. In contrast, thrombin-induced HPAEC cytoskeletal reorganization and barrier dysfunction are apparently dependent upon PKCδ-mediated CPI17 phosphorylation [105], suggesting that the involvement of CPI17 activation in EC barrier dysfunction is agonist-specific and depended upon EC origin. Bernal et al. demonstrated that under hypoxic conditions, a zinc-mediated increase in pulmonary EC contractility occurs via CPI17-dependent, but MYPT1-independent MLCP inhibition highlighting the role of CPI17 in MYPT1-independent MLCP regulation in ECs [106]. Accordingly, an increase in cAMP level by forskolin (FSK) activates the MLCP PPase activity via dephosphorylation and dissociation of the CPI17 in human umbilical vein ECs (HUVECs) [107]. Interestingly, aside from MLCP, several putative CPI17-interacting proteins were identified by two-hybrid screening of the lung library [108]. CPI17/plakoglobin (PG, γ-catenin) interaction was validated in HLMVECs by co-immunoprecipitation [108]. PMA-induced CPI17 phosphorylation decreases the association of CPI17 with plakoglobin, suggesting the involvement of CPI17 activation in the junctional reorganization of human microvascular ECs under edemagenic stimulation [108]. A role for CPI17 activation with targets other than MLCP family members upstream of RhoA was proposed in chicken amnion SM [109]. Accordingly, studies on proliferative cells demonstrated that CPI17 is involved in the regulation of histone phosphorylation independently from cytosolic action on MLCP [110]. The mechanisms of MLCP-independent CPI17 targeting remain to be determined.

Based on sequence homology, three other CPI17 family members (PHI-1 (Protein phosphate-induced 1 homolog), KEPI (kinase-enhanced PP1 inhibitor), and GBPI (gut and brain phosphatase inhibitor)) were identified (reviewed in [102]). They have more than 41% similarity in the central PHIN domain responsible for inhibiting PP1 holoenzyme and are capable of inhibiting MLCP [102]. PHI-1 is highly expressed in brain vascular endothelium and is involved in EC migration and retraction [111]. While the targeting PP1 for PHI-1 in ECs remains unknown, it suggests the involvement of PHI along with CPI17 in EC barrier regulation.

It has been known for years that MLC is the only substrate for MLCP. However, a growing body of evidence has demonstrated that MLCP via MYPT1 can target a variety of proteins involved in multiple cellular processes such as cell proliferation, division, development, and gene expression (reviewed in [81,89]). In particular, data from the literature revealed direct binding of MYPT1 with moesin/ezrin/radixin/moesin (ERM) family proteins and adducin [35,112,113]. These MYPT1-binding proteins are phosphorylated by Rho kinase and PKC and dephosphorylated by MLCP at conserved Thr near the C-terminus (Thr558 for moesin) [34,112,113,114,115,116]. ERM proteins act both as linkers between the actin cytoskeleton and plasma membrane proteins and as signal transducers in responses involving cytoskeletal remodeling (reviewed in [117,118,119]). In unphosphorylated form, ERM proteins exist in auto-inhibited conformation. Phosphorylation induces and stabilizes the unfolded active conformation of ERM proteins, allowing them simultaneously to bind to transmembrane receptors and actin filaments. The involvement of ERM proteins in EC barrier regulation is well-recognized [34,114,120,121,122]. In particular, it was shown that moesin phosphorylation is involved in EC barrier dysfunction in vitro and in vivo [34,123]. Interestingly, despite the structural similarity, ERM proteins are involved in EC barrier regulation in an agonist-specific manner. While all ERM proteins are involved in human lung EC barrier compromise, induced by TNFα [122] or microtubule disruption [114], moesin and ezrin promote EC barrier compromise and radixin opposes it in ECs stimulated by either thrombin or LPS [34,120]. It is worth noting that despite small structural differences, MYPT1 long and V2 are apparently distinct in their capabilities to bind ERM proteins [35]. The functional significance of this observation remains to be determined.

Adducin is a ubiquitously expressed membrane-skeletal protein, which in its unphosphorylated (active) form directly binds to and bundles actin filaments [113]. Consequently, adducin can modulate the lattice structure of the cytoskeleton and the exposure of transmembrane proteins (reviewed in [124,125]). Recent studies demonstrated that adducin is localized to tight junctions in response to shear stress and is involved in maintaining the blood–brain barrier [126]. In addition, thrombin induces adducin phosphorylation and adducin is directly involved in thrombin-induced human EC barrier compromise [127]. However, the role of adducin dephosphorylation in EC barrier regulation is undefined. Overall, the existence of multiple MYPT1 binding partners expanded our understanding of the MLCP function from a strictly myosin-targeting enzyme to a broader involvement of MLCP in the control of EC permeability.

Aside from MYPT1, several other members of the MYPT family were identified (reviewed in [91]). The PP1 regulatory subunit of the MYPT1 family, the TGF-β inhibited membrane-associated protein (TIMAP; PPP1R16B; GenBank ID#28626517), was first cloned in ECs and present in ECs in ample amounts [128]. Structural comparison with other MYPT family members reveals that, similar to MYPT1, TIMAP contains CS1 binding motif (RVxF), MyPhone motif, and ankyrin repeats in N-terminus (Figure 4). In contrast to MYPT1, TIMAP lacks inhibitory phosphorylation sites and Leucine zipper domains at the C-terminus but includes a CAAX box (prenylation site) at the C-terminus, allowing plasma membrane localization and additional regulatory phosphorylation sites for PKA/GSK3β (Glycogen synthase kinase-3β) and PKC (reviewed in [91,129]) (Figure 4). Similar to MYPT1, TIMAP binds with PP1_C_ β/δ isoform and MLC [53,130]. Further, it competes with MYPT1 for PP1_C_ [53], apparently decreasing MLCP activity towards MLC. However, does not preclude the possibility that TIMAP may direct PP1_C_ to other targets [53]. Importantly, TIMAP is down-regulated by edemagenic agonists, TGFβ and LPS, and protects EC barrier function in vitro and in vivo [128,130,131]. TIMAP interacts with moesin in pulmonary ECs and dephosphorylates (inhibits) it in a PKA-dependent manner [129]. In addition, TIMAP binds with scaffolding protein RACK1 (receptor for activated PKC), which is involved in cAMP/PKA-mediated EC barrier enhancement of pulmonary artery ECs [132]. Overall, data from the literature suggested that TIMAP functions are regulated via sequential PKA/GSK-3-mediated phosphorylation, which controls the TIMAP-RACK1 association and translocation of TIMAP to the cell membrane. Therefore, TIMAP targets PP1_C_ to the cell membrane and activates PP1-mediated moesin dephosphorylation, thus inhibiting its activation by edemagenic agonists such as thrombin [130,132,133,134]. In turn, PKCα binds and phosphorylates TIMAP at Ser331, resulting in reduced TIMAP/ERM protein binding and thus negatively affecting TIMAP/PP1_C_ activity towards ERM proteins [135]. In addition, a recent study [136] demonstrated that TIMAP interacts with annexin A2, a novel effector of EC barrier function, which interacts with VE-cadherin. TIMAP/PP1_C_ decreases PKC-mediated phosphorylation of annexin A2 at Ser25, which may affect EC barrier properties. It is also shown that for the TIMAP/PP1_C_ complex, elongation factor-1A1 acts as a novel substrate, which is involved in the regulation of EC spreading and attachment [137]. Interestingly, TIMAP may function independently in its role as a PP1 regulatory subunit [138,139], warranting the high versatility of TIMAP in regulating the EC cytoskeleton.

## 4. Ser/Thr Protein Phosphatase 2A

### 4.1. Structure-Functional Dynamics

Similar to PP1, the PP2A holoenzymes also occur in multimeric forms. Each of these holoenzymes has a common core which is derived from the scaffolding A subunit and the catalytic C subunit, which is associated with a diverse B subunit (regulatory) into a heterotrimeric complex [140,141] (Figure 1). More specifically, the core dimer consists of a 36 kDa common catalytic subunit (α or β), i.e., PP2A_C_ and 65 kDa structural or scaffolding subunit A (PP2A_A_; PR65), which is integrated with a third subunit (variable) known as regulatory subunit B (PP2A_B_). Scaffolding subunit A has two isoforms, namely α and β, consisting of 15 tandem huntingtin-elongation-A subunit TOR (HEAT) repeats that give U-shaped conformation. Each of these HEAT repeats is made of two antiparallel α-helices and comes with highly conserved interhelical loops. The conserved region within PR65 with exposed hydrophobic surfaces forms a binding site for the catalytic and any B subunit [140]. Structural studies evidenced that while HEAT repeats 1 to 10 (N-terminal) interact with a variety of regulatory subunits, HEAT repeats (11 to 15) tightly interact with the C subunit. Molecular dynamics studies evaluated that the A subunit has high flexibility that can adjust its shape (overall curvature) for binding a diverse range of partners [140]. Coordination by the A subunit results in the compilation of the catalytic subunit and any of the B subunits [140,142]. An analysis of the PP2A crystal structure revealed that PP2A_B_ interacts with PP2A_C_ near the active site, thus warranting substrate specificity [143]. The regulatory B subunit consists of four structurally unrelated gene families (B/B55/PR55, B′/B56/PR61, B″/PR72/PR70, and B‴/Striatin/PR93), comprising multiple isoforms and additional splice variants, which possess variable expression levels in various types of cells and tissues [51,140,141]. The regulatory B subunits give functional characteristics to PP2A by means of substrate specificity and altering catalytic activity [51,140,141]. The diverse functionality offered by the B subunit distinguishes a large number of heterotrimeric PP2A enzymes. The well-controlled process, a biosynthesis of active PP2A holoenzymes creates phosphatase specificity [144]. The unique structural diversity of B subunits allows the enzyme to control the majority of the subcellular compartments and drive a vast variety of cellular cascades including cell cycle, gene regulation, morphogenesis development, cell transformation, etc. (reviewed in [140,145,146,147,148,149]).

### 4.2. PP2A, Cytoskeleton, and EC Barrier Regulation

Though it is proven that PP2A is efficient towards isolated MLC, it does not effectively dephosphorylate the native myosin [150,151]. Accordingly, PP1, but not PP2A is the major enzyme responsible for MLC dephosphorylation in ECs [70]. Hence, as it does not play a direct catalytic role in the dephosphorylation of MLC, it was not considered a key enzyme in the regulation of contractility. However, a large body of evidence suggested its participation in the regulation of the cytoskeleton in various cell types (reviewed in [152,153,154]). In particular, PP2A is a well-recognized player in the regulation of MT organization [152,154]. Early studies demonstrated that PP2A is able to inhibit microtubule assembly by dephosphorylation of β_III_-tubulin [155]. Recently published data suggested that PP2A affects the cytoskeletal organization and drives dendrite diversity via targeting β-tubulin subunit 85D in Drosophila [156]. MT-mediated cell spreading is dependent upon the presence of the PP2A holoenzyme [157]. The regulatory B/PR55 subunit (α or β isoforms), but not members of the B’ family, co-sedimented with endogenous microtubules in the brain [158]. Accordingly, B/PR55, but not B΄/PR61, binding to PP2A_C_ is critical for cell survival under conditions of microtubule damage in yeast [159]. Interestingly, PP2A associates primarily with polymerized, but not monomeric, tubulin in the inactive state [152,160]. MT disruption releases PP2A, thus enhancing PP2A activity towards cytosolic tubulin and MT-associated proteins (MAPs) such as tau, MAP2, and MAP1B [152].

The MT-associated protein, tau, consists of several isoforms, formed by alternative mRNA splicing from a single gene [161]. While tau is predominantly found in neuronal cells, the expression of tau in non-neuronal cells, including endothelium, has also been reported [162,163,164]. Since the early 1990s, it was established that the capacity of tau to bind microtubules is regulated by phosphorylation. Unphosphorylated tau promotes MT assembly and inhibits the rate of depolymerization [165,166]. The phosphorylation of tau weakens its ability to bind microtubules and promote MT assembly [167,168,169]. While early studies on the role of tau dephosphorylation demonstrated that both PP2A and calcineurin (Ser/Thr phosphatase 2B) are involved in tau dephosphorylation, which restores its ability to promote MT assembly [170], a large body of evidence implicated PP2A encompassing B/PR55α subunit (PP2A/Bα) in tau dephosphorylation, leading to MT assembly [171]. This enzyme binds with tau via B/PR55α at the conserved domain with MT-binding repeats and is capable to dephosphorylate tau on several Ser/Thr phosphorylation sites in vitro and in vivo [171]. Similar to tau, PP2A is capable of dephosphorylating other MAPs, MAP1b and 2. The inhibition of PP2A leads to impairment of their MT-binding activity [172,173]. Apparently, MAP2 competes with tau for B/PR55α binding [174]. 

The involvement of microtubule remodeling in the regulation of lung EC permeability is well documented (reviewed in [5,7]). While disruption of the MT network resulted in EC barrier compromise [85,175,176], MT stabilization is involved in EC barrier protection in vitro and in vivo [177,178,179]. It was shown that PP2A is involved in MT-mediated human lung EC barrier regulation [37,164]. In quiescent ECs, PP2A co-localizes with microtubules. The pharmacologic inhibition of PP2A weakens the PP2A/MT association and potentiates the effect of a suboptimal dose of MT polymerization inhibitor, nocodazole, on TER [37]. Further, the ectopic expression of PP2A_A_ or PP2A_C_ opposes MT disruption, cytoskeletal remodeling, and an increase in EC permeability induced by microtubule inhibitor nocodazole or edemagenic agonists such as thrombin [164]. MT inhibitors or thrombin increased the phosphorylation of tau accompanied by tau translocation to the cell periphery and dissolution of the MT structure, suggesting the involvement of PP2A-mediated tau dephosphorylation in EC barrier regulation [37]. Accordingly, the ectopic expression of PP2A_A_ or PP2A_C_ attenuates thrombin- or nocodazole-induced phosphorylation of tau [164].

In addition, the ectopic expression of PP2A subunits attenuates the phosphorylation of small heat shock protein 27 (HSP27) [164]. HSP27 is a small chaperon, scaffolding, actin-capping protein, which is involved in protein folding, and regulation of the cytoskeleton (reviewed in [180,181]). Some data from the literature suggested that HSP27 associates with stable microtubules [182]. Further, specific inhibition of PP2A leads to the phosphorylation of HSP27, accompanied by MT destabilization and reorganization of the cytoskeleton [183]. HSP27 is phosphorylated by MAP kinase-activated protein kinase 2 (MAPKAP kinase 2), which is, in turn, phosphorylated and activated by p38 MAP kinase [184,185,186,187]. The phosphorylation of HSP27 promotes F-actin formation, and membrane blebbing, and mediates actin reorganization and cell migration in human endothelium [184,185,188,189]. The role of HSP27 in EC barrier regulation is complex and may be agonist-specific [190,191,192,193,194]. In sum, data from the literature suggest that HSP27 may be involved in the interplay between the microtubule and actin cytoskeleton and the phosphorylation status of this protein may be important for the regulation of EC cytoskeletal organization and permeability.

While PP2A is capable of dephosphorylating HSP27 in some cell types [195,196], an early study demonstrated the association of phosphorylated MYPT1 with HSP27 in SM cells, which coincides with MLCP inhibition [197]. The inhibition of PP1 and 2A by calyculin A blocks HSP27 dephosphorylation induced by ischemic pre-conditioning of cardiomyocytes [198].

Whether PP2A or PP1 (or both) are capable to dephosphorylate HSP27 thus regulating EC permeability remains to be determined.

While human lung ECs express many isoforms of PP2A regulatory subunits (Figure 5), to date only the involvement of the Bα subunit in the regulation of EC permeability was documented [36]. Similar to pharmacologic PP2A inhibition, the depletion of Bα results in MT dissolution and actin remodeling. Further, it delays EC barrier recovery after thrombin insult. PP2A pharmacologic inhibition or depletion of Bα increases phosphorylation of the junctional protein, β-catenin at Ser552 accompanied by its translocation from membrane to the cytoplasm, suggesting the involvement of PP2A holoenzyme A/Bα/C in EC barrier regulation through the remodeling of adherens junctions (AJs). Aside from ECs, the multifaceted involvement of PP2A in the junctional assembly was reported in several cell types, primarily in the epithelium [199,200], reviewed in [201]. However, the direct involvement of PP2A-mediated dephosphorylation of β-catenin at Ser552 in AJ assembly and EC barrier regulation requires further investigation.

Bα is required for maintaining MT stability and endothelial tube formation in HUVECs and vessel lumen integrity in zebrafish [202]. It was proposed that PP2A inhibits the RhoA/Rho kinase/MLC pro-contractile pathway by suppressing transcriptional activation of the adaptor protein, Arg-binding protein 2 (ArgBP2), via binding and dephosphorylation of its transcriptional co-repressor histone deacetylase 7 (HDAC7). The PP2A holoenzyme A/Bα/C directly binds and dephosphorylates HDAC7, thus warranting nuclear localization of HDAC7 and repression of the ArgBP2 gene [202]. HDAC7 belongs to the class IIa family of HDACs, which are capable of shuttling between the nucleus and cytoplasm. Nuclear export leads to their dissociation from nuclear targets, thus abrogating their roles as co-repressors of transcription (reviewed in [203,204]). Recently, the protective role of class IIa HDACs inhibition in LPS-induced lung ECs barrier compromise in vitro and in vivo was demonstrated [205]. Interestingly, the LPS effect on the EC barrier was attributed, at least in part, to an increase in ArgBP2 expression, which can be downregulated by class II HDAC inhibitors. These data suggested that PP2A-mediated inhibition of HDAC-mediated transcriptional activity may be involved in EC barrier regulation. However, the exact link between PP2A activity, suppression of HDAC-mediated ArgBP2 expression, and EC permeability remains to be determined.

In addition, the data from Young et al. (2002) revealed that PP2A may regulate the motility of ECs by controlling the stability of focal adhesion complexes (FAs) via the modulation of Ser/Thr paxillin phosphorylation [206]. Paxillin is a main component of focal adhesions and is involved in multiple mechano-transduction events (reviewed in [207]). The phosphorylation of paxillin at various Ser/Thr residues plays a role in the paxillin-mediated assembly/disassembly of Fas (reviewed in [208]). The phosphorylation of paxillin by p21-activated kinase-1 (PAK1) at Ser273 may contribute to Rac/PAK1-mediated sustained barrier strengthening induced by oxidized phospholipids [209]. A recent study demonstrated that PP2A inhibition decreases paxillin interaction with FA kinase (FAK) in NIH3T3 cells, thus reducing the formation of FAs [210]. However, the involvement of PP2A-mediated paxillin dephosphorylation in the regulation of the EC barrier requires further investigation. 

Recent genetic work on flies demonstrated that the PP2A B subunit, *twins,* may be involved in actin polymerization in vivo [211]. However, the role of PP2A-mediated actin polymerization in EC barrier regulation remains to be determined. Overall, the data from the literature supported the multifaceted involvement of PP2A in cytoskeletal organization.

### 4.3. Crosstalk between PP2A and MLCP in the Regulation of EC Barrier Function

The coordinated involvement of PP2A and MLCP in EC barrier regulation became evident in the role of these enzymes in the regulation of endothelial nitric oxide synthase (eNOS). Guanosine 3′,5′-cyclic monophosphate (cGMP) and nitric oxide (NO) are the key players in the regulation of endothelial barrier function [212,213,214,215]. The involvement of NO/eNOS in vascular permeability is highly complex and agonist- and tissue-specific. While a normal NO level is required to maintain vascular function, the hyperactivation of eNOS increases NO production, leading to microvascular EC permeability in response to inflammatory agonists such as vascular endothelial growth factor (VEGF) and histamine (reviewed in [216,217,218]). The importance of the EC cytoskeleton and reversible phosphorylation in the regulation of NO-mediated EC barrier function has been documented [216,218,219,220,221]. The phosphorylation/dephosphorylation of eNOS at Ser1177 and Thr495 (human sites) is an important determinant of eNOS activity [220,222]. While in unstimulated conditions Thr495 is phosphorylated, the level of Ser1117 phosphorylation is low [222]. Pro-inflammatory agonists such as VEGF activate eNOS via Akt (protein kinase B)-mediated phosphorylation at Ser1177 [223,224]. In turn, Rho kinase and PKC phosphorylate eNOS at Thr495 [225,226]. Greif et al. (2002) demonstrated that PP2A is involved in the dephosphorylation of both Ser1177 and Thr495. The pharmacologic inhibition of PP2A increases eNOS activity [227]. Further, some data from the literature suggested that the phosphorylation at Thr495 prevents Ser1177 phosphorylation, thus decreasing eNOS activity [227]. However, a recent study identified MLCP as an enzyme directly involved in eNOS dephosphorylation at Thr497 (Thr495 in humans) in bovine pulmonary artery ECs (BPAECs) [228]. MLCP interacts with eNOS via MYPT1. PKA-mediated PP2A activation induced by green tea polyphenol, epigallocatechin gallate (EGCG), leads to MYPT1 phosphorylation, thus promoting MLCP activation (de-inhibition) resulting in the dephosphorylation of eNOS at Thr497 and dephosphorylation of MLC at Thr18/Ser19 [228]. Overall, these data support the crosstalk between PP2A and MLCP in regulating eNOS activity and EC contractility (Figure 6).

In addition to dephosphorylating MYPT1, PP2A is capable of dephosphorylating the MLCP inhibitor, CPI17 [229] (Figure 6). In contrast to MYPT1 dephosphorylation, the rate of CPI17 dephosphorylation was comparable to the rate of MLC dephosphorylation in SM, suggesting that PP2A-mediated CPI17 dephosphorylation can efficiently regulate SM contractility [229]. Activated (phosphorylated) CPI17 is involved in the regulation of the EC cytoskeleton and permeability, and SM contractility via direct binding and inhibition of MLCP (see Section 3.2 above for more details). Therefore, PP2A may activate (de-inhibit) MLCP activity independently of MYPT1 phosphorylation (Figure 6). Further, published data [108] demonstrated that CPI17 is the plakoglobin binding partner suggesting the involvement of CPI17 in AJ formation in the endothelium. PP2A may be involved in AJ remodeling via the dephosphorylation of β-catenin [36]. However, the link between PP2A-mediated CPI17 dephosphorylation, AJs assembly, and EC barrier regulation remains to be determined.

## 5. Conclusions

Reversible Ser/Thr phosphorylation is a major post-translational modification orchestrating a diverse array of protein–protein interactions and cellular signaling pathways. Ser/Thr protein kinases and protein phosphatases account for the vast majority of phosphorylation events and are ultimately involved in regulating critical cellular processes, including the regulation of the EC barrier. A considerable body of evidence now supports the involvement of major Ser/Thr phosphatases of the PPP family, PP1 and PP2A, in the regulation of EC contractility, cytoskeletal remodeling, and barrier regulation via the dephosphorylation of specific cytoskeletal targets and regulatory proteins. Further, there is evidence of the interplay between PP1 and PP2A in the regulation of endothelial permeability. However, a more complete understanding of phosphatase-mediated cellular processes and the role of specific phosphatase complexes in agonist-induced EC cytoskeletal remodeling and permeability regulation remains to be defined. 

While developing new therapeutic approaches based on PP1/PP2A targeting is a promising direction in the treatment of ALI/ARDS, it is a highly challenging area of research due to broad substrate specificity along with the high structural similarity of the free catalytic subunits and high complexity of protein interactomes of PPP family PPases (i.e., PP1 and 2A) [39,40,41]. According to recent data from the literature (reviewed in [69,230]), one of the methods to modulate PPase complexes is based upon targeting the disruption of the existing harmful complexes, thus liberating catalytic subunits, which can de-phosphorylate other substrates in close proximity. In this regard, the development of small molecule inhibitors, which can compete with specific docking motifs of PP1/PP2A regulatory subunits thus disrupting selective PPase interactomes, may open a new avenue in the treatment of cardiovascular and pulmonary diseases.

## Figures and Tables

**Figure 1 biomedicines-11-01638-f001:**
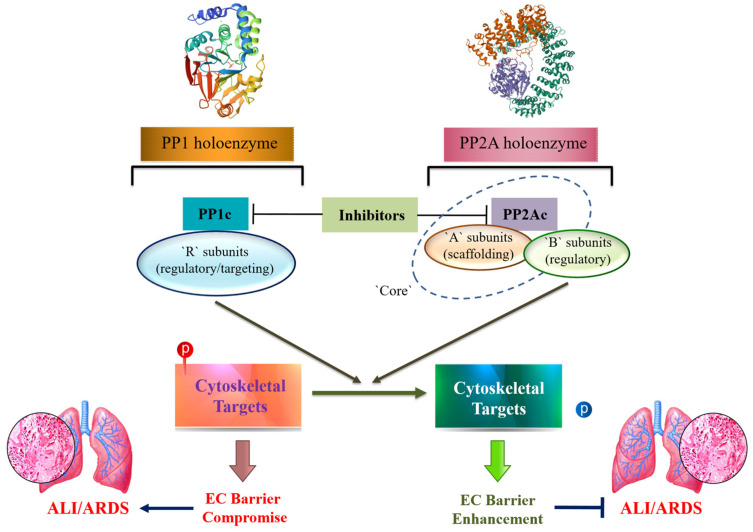
Simplified presentation of structural features and functional roles of PP1 and PP2A in lung EC barrier regulation. Insights are provided in the text. PP1: Protein phosphatase 1; PP2A: Protein phosphatase 2A; PP1_C_ and PP2A_C_: catalytic subunits of PP1 and PP2A, respectively; EC: Endothelial cell; ALI: Acute lung injury; ARDS: Acute respiratory distress syndrome. PubMed structure IDs: 24591642 (PP1); 17086192 (PP2A).

**Figure 2 biomedicines-11-01638-f002:**
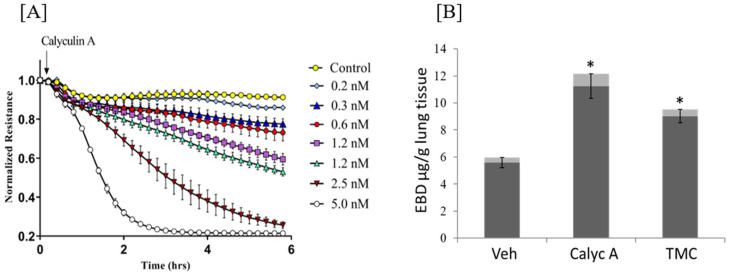
Inhibition of protein phosphatase 1 (PP1) and protein phosphatase 2A (PP2A) compromises the endothelial barrier in vitro and in vivo: (**A**) Calyculin A (added at arrow) decreases transendothelial electrical resistance (TER) of human lung microvascular ECs (HLMVECs) in a dose-dependent manner; (**B**) Calyculin A (Calyc A) and tautomycetin (TMC) given intravenously (i.v. 10 nM in blood final concentration or 0.4 µg/mg of lung tissue, respectively), increases lung vascular leak measured by BSA-Evans-Blue dye (EBD) extravasation. N ≥ 3, * *p* < 0.05. Veh: Vehicle. Unpublished data from the laboratory of Dr. Verin.

**Figure 3 biomedicines-11-01638-f003:**
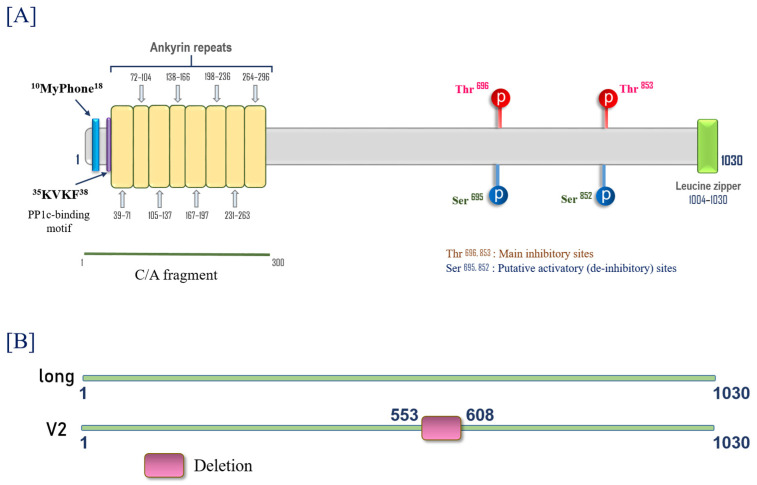
(**A**) The structure of human endothelial cell myosin phosphatase targeting subunit 1 (EC MYPT1). The positions of main functional domains and regulatory phosphorylation sites were determined based on human EC MYPT1 sequence [35] and data from the literature; (**B**) The schematic representation of the structural differences between two EC MYPT1 isoforms [35].

**Figure 4 biomedicines-11-01638-f004:**
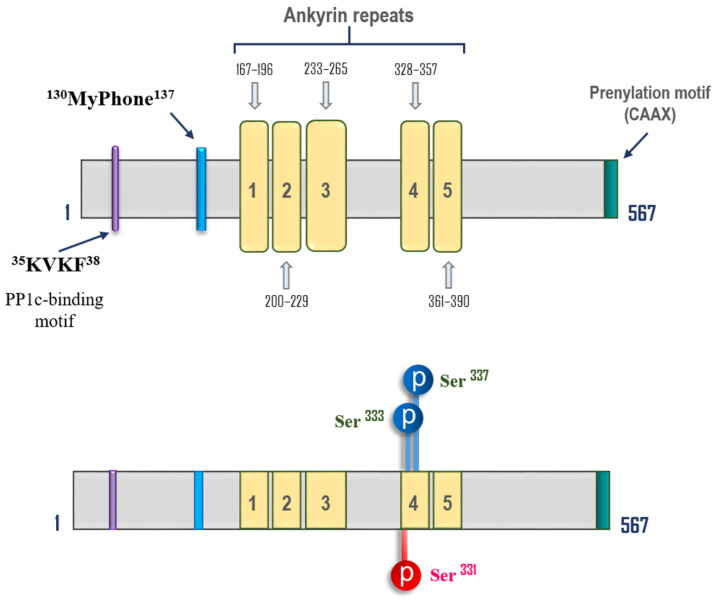
The structure of human TGF-β inhibited membrane-associated protein (TIMAP). The positions of the main functional domains (upper panel) and regulatory phosphorylation sites (lower panel) are shown based on the human TIMAP sequence and data from the literature [129]. The activatory PKA/GSK3β phosphorylation sites are shown in blue; the inhibitory PKC site is shown in red.

**Figure 5 biomedicines-11-01638-f005:**
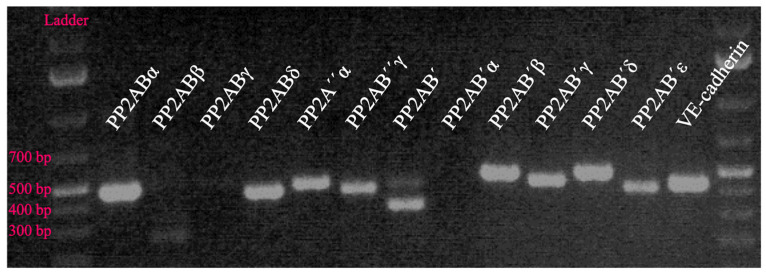
Human lung microvascular ECs (HLMVECs) express the majority of known B subunits of protein phosphatase 2A (PP2A). Shown is an expression profile obtained by polymerase chain reaction (PCR) analysis. The level of vascular endothelial (VE)-cadherin expression (specific for ECs) is shown for comparison. Specific primers are shown in Appendix A. Unpublished data from the laboratory of Dr. Verin.

**Figure 6 biomedicines-11-01638-f006:**
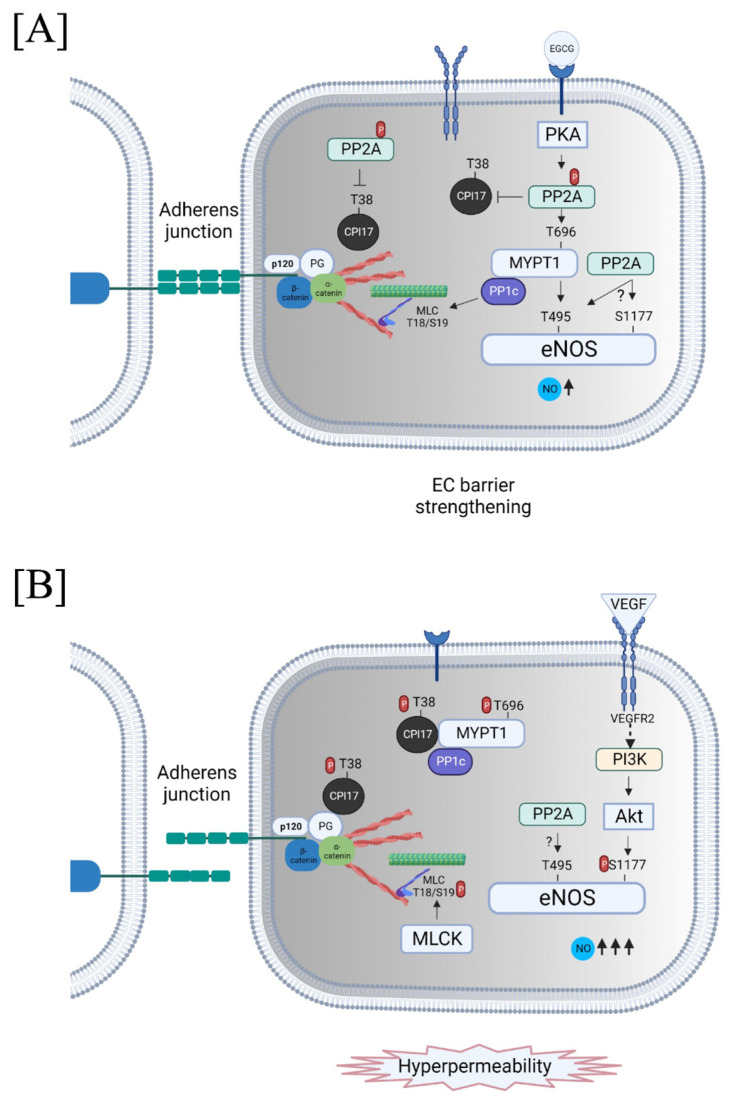
Crosstalk between protein phosphatase 2A (PP2A) and myosin light chain (MLC) phosphatase (MLCP) in pulmonary EC barrier regulation. (**A**) PKA-mediated activation of PP2A strengthens the EC barrier via dephosphorylation (deactivation) of specific MLCP inhibitor, CPI17, and dephosphorylation of Myosin phosphatase targeting subunit 1 (MYPT1) leading to MLCP activation (de-inhibition of PP1c catalytic activity towards MLC), which may be accompanied by direct PP2A-mediated endothelial nitric oxide synthase (eNOS) dephosphorylation. (**B**) Pro-inflammatory agonists such as VEGF may hyperactivate eNOS leading to excessive NO production. In parallel, pro-inflammatory agonists may inhibit MLCP via direct phosphorylation of MYPT1 resulting in the inhibition of PP1c catalytic activity toward MLC or activation (phosphorylation) of CPI17 leading to EC hyperpermeability. Additional abbreviations: EGCG—epigallocatechin gallate; VEGFR2—VEGF receptor 2; PG—plakoglobin (γ-catenin). This figure was created with BioRender.com.

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
