# Peer review of "Serine/Threonine Protein Phosphatases 1 and 2A in Lung Endothelial Barrier Regulation"

_biomedicines, 2023, doi:10.3390/biomedicines11061638_

Round 1

Reviewer 1 Report

In the manuscript, “Serine/Threonine Protein Phosphatases 1 and 2A in Lung Endothelial Barrier Regulation,” the authors review the current understanding of serine/threonine phosphoprotein phosphatases in the regulation of lung endothelial barrier function. The authors describe a) the structure and functional dynamics of both PP1 and PP2a, b) the current evidence to support PP1 and PP2a in regulating endothelial barrier function, and c) the crosstalk between protein phosphatases and important cytoskeletal regulators such as MLCP. The authors conclude that serine/threonine protein phosphatases have important contributions to endothelial cell contractility, cytoskeletal remodeling, and barrier function.

The manuscript sheds important light on an understudied area in pulmonary vascular biology. The authors provide a comprehensive review detailing the complex mechanisms involved in dephosphorylation of certain cytoskeletal elements that are important to endothelial barrier dysfunction. Given the complex mechanisms described, I would like to see the authors focus on improving the clarity, particularly given that a wide reaching audience for this journal is likely not at the expert level of the authors on this topic. I have included some comments below for the authors to consider.

Comments

1.    Please be sure to define all acronyms, MYPT1 is listed in line 170, but not properly described or defined prior to this. Similarly, MLCP is not defined in line 183. Please review manuscript for other acronyms that are not well described or defined.

2.    The mechanisms discussed regarding PP1 and endothelial barrier function are very complex and the description can be easily confusing at times, I think it would be important to consider a figure to better describe the different pathways involved.

3.    Additionally, the description of the mechanisms in section 3.2 PP1 and the regulation of endothelial barrier function from lines 183 to 310 have very little discussion specifically of PP1, while it is clear that phosphorylation states of these cytoskeletal components is important to EC barrier function, it is more important to understand where PP1 fits into this schema. 

4.    It is unclear if the data presented in Figure 1 and Figure 4 is the authors unique data or if these are figures pulled from other published works, in which case a reference should be included and any copyright issues should be addressed / listed.

5.    On page 10, there is discussion of HSP27 in lines 391-397, but the description of the importance of this protein to the cellular cytoskeleton is not mentioned until the next paragraph. This seems out of order and the discussion in multiple different cell types in lines 391-397 seems unnecessary for this review of endothelial cell function.

6.    In Figure 5, PP1c is not mentioned in the caption, but given that this is one focus of the review paper, it would be important to comment on the of PP1c in this schema.

7.  All acronyms used in figures should be defined in the figure legend/caption.

Author Response

Thank you for the valuable suggestions and critical review. Please validate the response to the comments.

Reviewer 1:

Comments

  1. Please be sure to define all acronyms, MYPT1 is listed in line 170, but not properly described or defined prior to this. Similarly, MLCP is not defined in line 183. Please review the manuscript for other acronyms that are not well described or defined.

Ans. The changes were implemented as per the suggestions. Also, the entire manuscript is reviewed for the acronyms.

  1. The mechanisms discussed regarding PP1 and endothelial barrier function are very complex and the description can be easily confusing at times, I think it would be important to consider a figure to better describe the different pathways involved.

Ans. Thank you for suggestion. While it is hard to combine all PP1-mediated signaling in one figure due to complexity, we essentially extended and clarified the description of signaling in the text. In addition, we generated new Figure 1 with simplified schema illustrating structural features and functional roles of PP1 and PP2A in endothelial barrier regulation.

  1. Additionally, the description of the mechanisms in section 3.2 PP1 and the regulation of endothelial barrier function from lines 183 to 310 have very little discussion specifically of PP1, while it is clear that phosphorylation states of these cytoskeletal components is important to EC barrier function, it is more important to understand where PP1 fits into this schema.

Ans. Section 3.2 was significantly expanded according to the reviewer’s suggestions.  

  1. It is unclear if the data presented in Figure 1 and Figure 4 is the authors unique data or if these are figures pulled from other published works, in which case a reference should be included and any copyright issues should be addressed / listed.

Ans. These are unpublished data from Dr. Verin’s laboratory. This information was added to the figure legends.

  1. On page 10, there is discussion of HSP27 in lines 391-397, but the description of the importance of this protein to the cellular cytoskeleton is not mentioned until the next paragraph. This seems out of order and the discussion in multiple different cell types in lines 391-397 seems unnecessary for this review of endothelial cell function.

Ans. The discussion on the role of HSP27 has been revised accordingly.

  1. In Figure 5, PP1c is not mentioned in the caption, but given that this is one focus of the review paper, it would be important to comment on the of PP1c in this schema.

Ans. The role of PP1c is mentioned in the caption.

  1. All acronyms used in figures should be defined in the figure legend/caption.

Ans. All acronyms in the figures are defined in the legends.

Reviewer 2 Report

Biomedicines (ISSN 2227-9059)

biomedicines-2406392

Review

Serine/Threonine Protein Phosphatases 1 and 2A in Lung Endothelial Barrier Regulation

Patil et al.

Premise: Endothelial barrier dysfunction is an important contributor to acute respiratory failure, and the role of Ser/Thr phosphorylation of cytoskeletal components is well documented. Much less is known about the counterregulatory effects of Ser/Thr phosphatases in endothelial barrier function. This review will specifically focus on endothelial phosphatases 1and 2A, since they are most abundant and responsible for the majority of dephosphorylation events in the lung.

Strengths:

1-    Endothelial barrier dysfunction is an important and timely topic as a key contributor to acute respiratory failure/ARDS.

2-    No targeted therapeutic approaches are available against alveolar-capillary barrier dysfunction that have translated into improved patient outcomes.

Weaknesses/areas for improvement:

1-    Highly technical content that limits direct applicability to a larger audience interested in ALI/ARDS. Maybe inserting an early diagram with a general overview and break-down of the pathways that will be discussed could prepare the reader better and give a more easily digestible framework for the highly technical pages to follow.

2-    It is a bit unclear why sections 3 (PP1) and 4 (PP2A) do not have the same structure/headings since the manuscript focus on these two phosphatases. Aligning them better would improve the overall clarity.

3-    While there is no issue with focusing solely on endothelial barrier function for this review, the fact that most of the barrier tightness in the alveolar-capillary unit is provided by the epithelium, not the endothelium, should be addressed upfront with the rationale for focusing on just the endothelium.

4-    It would be helpful if the review concluded with a section of where to go from here, and by pointing out how all this detailed knowledge could be translated into clinically-relevant targeted therapeutic approaches. What models could be used? What will be foreseeable challenges?

Author Response

Thank you for the valuable suggestions and critical review. Please validate the response to the comments.

Reviewer 2:

Comments

  1. Highly technical content that limits direct applicability to a larger audience interested in ALI/ARDS. Maybe inserting an early diagram with a general overview and breakdown of the pathways that will be discussed could prepare the reader better and give a more easily digestible framework for the highly technical pages to follow.

Ans. A simplified schema was added to the text. In addition, we substantially revised the text for clarity with additional details explaining the role of distinct signaling pathways in endothelial barrier regulation.

  1. It is a bit unclear why sections 3 (PP1) and 4 (PP2A) do not have the same structure/headings since the manuscript focus on these two phosphatases. Aligning them better would improve the overall clarity.

Ans. We adjusted headings accordingly as the reviewer suggested.

  1. While there is no issue with focusing solely on endothelial barrier function for this review, the fact that most of the barrier tightness in the alveolar-capillary unit is provided by the epithelium, not the endothelium, should be addressed upfront with the rationale for focusing on just the endothelium.

Ans. As the reviser correctly mentioned, the manuscript is focused entirely on the role of phosphatases in endothelial, but not epithelial barrier regulation. The regulation of epithelial barrier is different and out of scope of this review. However, we now acknowledged the role of epithelial barrier in the alveolar-capillary injury in ALI in the Introduction.   

  1. It would be helpful if the review concluded with a section of where to go from here, and by pointing out how all this detailed knowledge could be translated into clinically-relevant targeted therapeutic approaches. What models could be used? What will be foreseeable challenges?

Ans. This section was added to the manuscript.

Round 2

Reviewer 1 Report

The authors have now made appropriate modifications to the manuscript and I have no further comments to add.